# The Role of Interleukin-10 in the Pathogenesis and Treatment of a Spinal Cord Injury

**DOI:** 10.3390/diagnostics14020151

**Published:** 2024-01-09

**Authors:** Christos Patilas, Iordanis Varsamos, Athanasios Galanis, Michail Vavourakis, Dimitrios Zachariou, Vasileios Marougklianis, Ioannis Kolovos, Georgios Tsalimas, Panagiotis Karampinas, Angelos Kaspiris, John Vlamis, Spiros Pneumaticos

**Affiliations:** 3rd Department of Orthopaedic Surgery, National & Kapodistrian University of Athens, KAT General Hospital, 14561 Athens, Greece

**Keywords:** spinal cord, spinal cord injury treatment, interleukin-10, review

## Abstract

Spinal cord injury (SCI) is a devastating condition that often leads to severe and permanent neurological deficits. The complex pathophysiology of an SCI involves a cascade of events, including inflammation, oxidative stress, and secondary injury processes. Among the myriad of molecular players involved, interleukin-10 (IL-10) emerges as a key regulator with the potential to modulate both the inflammatory response and promote neuroprotection. This comprehensive review delves into the intricate interplay of IL-10 in the pathogenesis of an SCI and explores its therapeutic implications in the quest for effective treatments. IL-10 has been found to regulate inflammation, oxidative stress, neuronal apoptosis, and glial scars after an SCI. Its neuroprotective properties have been evaluated in a plethora of animal studies. IL-10 administration, either isolated or in combination with other molecules or biomaterials, has shown neuroprotective effects through a reduction in inflammation, the promotion of tissue repair and regeneration, the modulation of glial scar formation, and improved functional outcomes. In conclusion, IL-10 emerges as a pivotal player in the pathogenesis and treatment of SCIs. Its multifaceted role in modulating inflammation, oxidative stress, neuronal apoptosis, glial scars, and neuroprotection positions IL-10 as a promising therapeutic target. The ongoing research exploring various strategies for harnessing the potential of IL-10 offers hope for the development of effective treatments that could significantly improve outcomes for individuals suffering from spinal cord injuries. As our understanding of IL-10′s intricacies deepens, it opens new avenues for innovative and targeted therapeutic interventions, bringing us closer to the goal of alleviating the profound impact of SCIs on patients’ lives.

## 1. Introduction

The spinal cord is a delicate structure encased in the vertebral column, responsible for transmitting signals between the brain and the rest of the body. Spinal cord injury (SCI) represents a significant public health concern, affecting millions of individuals worldwide [1]. An SCI can result from traumatic events such as motor vehicle accidents, falls, or sports injuries, leading to immediate damage (primary injury) and subsequent cascades of cellular and molecular events (secondary injury) that contribute to further neurological deterioration [2]. Concerning the primary injury, the force applied to the spinal cord can cause fractures or dislocations of the vertebrae, leading to compression or contusion of the spinal cord. Mechanical trauma can cause immediate vascular damage, leading to hemorrhage and ischemia. Disruption of blood vessels reduces oxygen and nutrient supply to spinal cord tissues, exacerbating cell damage [3,4].

While the primary injury sets the stage, secondary injury mechanisms significantly contribute to the progression of tissue damage in SCIs. Within hours of the primary injury, a robust inflammatory response is triggered. Activated immune cells, including microglia and infiltrating macrophages, release pro-inflammatory cytokines (TNF-α, IL-1β, and IL-6) and chemokines, contributing to neuroinflammation [5,6]. The inflammatory cascade not only exacerbates tissue damage but also creates a hostile microenvironment that hinders regenerative processes. Glutamate, a neurotransmitter, is released excessively after SCIs. This glutamate surplus leads to excitotoxicity, causing an influx of calcium ions into neurons and subsequent cell death. Excitotoxicity contributes to the loss of neurons and disrupts the delicate balance of neurotransmission [7]. The inflammatory response and disrupted blood supply result in the production of reactive oxygen species (ROS). ROS, in excess, induces oxidative stress, causing damage to lipids, proteins, and DNA. Oxidative stress amplifies neurodegenerative processes, perpetuating the cycle of cell death and tissue damage [8]. Moreover, after SCIs, the integrity of the blood–brain barrier is compromised, allowing the infiltration of immune cells, proteins, and other potentially harmful substances into the spinal cord. Blood–brain barrier disruption further amplifies inflammation and contributes to the secondary injury cascade. Both neuronal and glial cell apoptosis and necrosis contribute to the loss of neural tissue and exacerbate the neurological deficits associated with SCIs [9]. Wallerian degeneration, the progressive disintegration of the axon, occurs distal to the injury site. Axonal injury contributes to the disruption of neural circuits, hindering communication between the brain and the rest of the body [10]. Finally, glial cells, including astrocytes and microglia, become activated in response to SCIs. While they play a role in tissue repair, their activation can lead to the formation of a glial scar, which creates a physical and chemical barrier to regeneration [11]. However, this fibrotic barrier impeded axonal regeneration and recruited immune cells, making for delayed injury in the chronic phase [12]. The balance between the beneficial and detrimental effects of glial activation is crucial for functional recovery.

Inflammation plays a pivotal role in the secondary injury phase of SCIs. The release of pro-inflammatory cytokines, chemokines, and ROS contributes to the recruitment of immune cells, glial activation, cleaning of cellular debris, and neuronal apoptosis. While inflammation is a natural response to injury, an uncontrolled and prolonged inflammatory cascade exacerbates tissue damage, hindering the regenerative capacity of the spinal cord [13]. The bone-marrow-derived cells that are stimulated with LPS or interferon-γ and depend on STAT1 to produce tumor necrosis factor-α, IL-1, and nitric oxide synthase (previously known as M1 type of microglia/macrophages) may exacerbate secondary tissue damage, whereas the cells that are stimulated by IL-4, rely on STAT6, produce arginase, and augment Mrc and Ym1 (previously known as M2 type) may trigger and promote reparative tissue repair. Beyond the immediate inflammatory response, an ongoing release of cytokines persists in the chronic phase of SCIs. This chronic inflammation contributes to secondary injury processes, hindering the potential for neural regeneration [14,15,16].

Interleukin-10 (IL-10), initially identified as a cytokine with potent anti-inflammatory properties, is produced by various immune cells, including macrophages, T cells, and B cells. IL-10 is a symmetric homodimer that exerts its effects by binding to a cell surface receptor, the interleukin-10 receptor (IL-10R) [17,18]. The binding of IL-10 to its receptor initiates a series of intracellular signaling events, activating the JAK–STAT pathway [19]. Activated JAKs phosphorylate tyrosine residues on the receptor subunits, creating docking sites for STAT proteins. Phosphorylated STAT3 proteins form dimers and translocate to the nucleus, where they act as transcription factors. Within the nucleus, STAT3 dimers bind to specific DNA sequences in the promoter regions of target genes, regulating their transcription. Its primary function is to suppress the synthesis and release of pro-inflammatory cytokines, modulate immune cell function, and promote an anti-inflammatory microenvironment [20].

Dysregulation of IL-10 has been implicated in autoimmune diseases such as rheumatoid arthritis, inflammatory bowel disease, and multiple sclerosis [21]. Deficient IL-10 responses may contribute to the perpetuation of chronic inflammation and autoimmunity. IL-10 plays a dual role in infectious diseases. While it contributes to the resolution of inflammation and prevents tissue damage during infections, excessive IL-10 production can impair the immune response, facilitating pathogen persistence. Immunoregulatory functions of IL-10 have implications for cancer immunology and the pathogenesis of neuroinflammation and neurological disorders. Its anti-inflammatory properties may have implications for conditions such as Alzheimer’s disease, Parkinson’s disease, and traumatic brain injury [22,23,24].

Modulating IL-10 responses in the central nervous system is an area of active research for potential therapeutic interventions.

## 2. Relevant Sections

### 2.1. Detection of IL-10 in Spinal Cord Injury

Although IL-10 was found to be produced by various immune cells, including macrophages, T cells, and B cells, its secretion was detected in several CNS cells, such as microglia, activated astrocytes, macrophages/monocytes, oligodendrocytes, and neurons [25,26]. IL-10 has been detected in the cerebrospinal fluid, which surrounds the spinal cord. Moreover, IL-10R has been detected in microglia, astrocytes, oligodendrocytes, and neurons, as well as embryonic spinal cord neurons [27]. IL-10 attachment to the high-affinity IL-10R by paracrine and autocrine interactions modulates glia-mediated inflammatory responses [25].

In vivo and in vitro studies have observed that, in the acute phase of SCIs, IL-10 along with IL-10R respond quickly [26,27,28,29]. In vitro studies have shown that the levels of IL-10 mRNA peaked around 1 h after an SCI and returned to basal levels 7 days post-SCI [28,30,31]. IL-10 protein production was significantly increased at 24 h after an SCI and peaked at approximately 1 or 2 weeks after an SCI [28,32,33]. On the other hand, the expression of IL-10 mRNA and IL-10 protein remained low in the chronic phase of an SCI [25,34]. IL-10 may be regarded as a serologic marker to forecast an SCI, and high serum levels of IL-10 may indicate a better prognosis [35]. As age increases, the production of IL-10 after an SCI is reduced [36]. The brief increase in IL-10 and its receptor during the primary SCI and the subsequent decrease at the secondary phase raise the possibility that exogenous supplementation of IL-10 may suppress secondary injury and limit the damage to the SCI.

### 2.2. IL-10 in the Pathogenesis of Spinal Cord Injury

#### 2.2.1. Regulation of Inflammation

One of the most important roles of IL-10 in an SCI is the suppression of secondary inflammation. IL-10 exerts its anti-inflammatory effects through various mechanisms. It inhibits the production of pro-inflammatory cytokines, such as TNF-α, IL-1β, IL-6, and IL-12, which significantly improved functional recovery following traumatic SCIs in rats [33,37,38,39,40]. After an SCI, IL-10 decreases NOS release from microglia [40]. Neutrophil activity, along with TNF-α and IL-1β, was significantly augmented in spinal cord sections of IL-10-deficient mice after an SCI [32]. IL-10 can impact the expression of chemokines, such as CXCL5, which are molecules involved in the recruitment of immune cells to the site of injury [38]. By regulating chemokine expression, IL-10 helps control the influx of immune cells and modulates the local immune environment [41,42].

IL-10 inhibits the activation and proliferation of T cells, including both CD4+ and CD8+ T cells. By limiting T cell responses, IL-10 helps prevent excessive immune activation and the release of pro-inflammatory factors that can contribute to secondary injury [43]. However, IL-10 can induce regulatory T cells, leading to suppression of microglia activation, reduction in recruitment of peripheral monocytes, stabilization of local inflammatory storms, and reduction in neurodegeneration [44].

Additionally, IL-10 suppresses the activation of immune cells, including macrophages and microglia, preventing their pro-inflammatory polarization, which means that IL-10 can even play a therapeutic role in the chronic phase [32]. Administration of IL-10 in SCI rats resulted in significantly less M1 cells promoting the M2 macrophage/microglia phenotype, enhancing axonal regrowth, neural regeneration, remyelination, and functional recovery [39,45]. M2 macrophages have been found to produce higher levels of IL-10 at injured spinal segments, leading to less spinal cord lesion volume, increased axonal myelination, and neuronal preservation, resulting in improved locomotor function [26]. IL-10-deficient monocytes failed to promote recovery, suggesting that IL-10 is an important factor for the beneficial function of monocyte-derived macrophages in spinal cord recovery [46].

#### 2.2.2. Regulation of Oxidative Stress

The specific mechanisms by which IL-10 influences oxidative stress after an SCI may vary. It is known that overexpression of iNOS can lead to excessive production of NO. Excessive NO production, particularly when combined with an increase in superoxide radicals, can contribute to oxidative stress. IL-10 downregulates the expression of iNOS protein and mRNA, preventing neuronal apoptosis and improving function [40]. In SCI mice, IL-10 administration greatly suppressed iNOS production [45]. In spinal cord sections of IL-10-deficient mice, iNOS expression was significantly increased [32]. Animal studies have suggested an important role for the IL-10/miR-155 pathway in regulating NADPH oxidase-mediated SCI dysfunction. Depletion of NADPH oxidase promotes IL-10 expression following an SCI, contributing to improved functional recovery [47]. Taking these into consideration, IL-10 may prevent ROS-induced damage to the spinal cord, suggesting that IL-10 administration may have beneficial effects on SCIs.

#### 2.2.3. Regulation of Apoptosis

IL-10 has been shown to have anti-apoptotic properties through the JAK–STAT pathway in various cellular contexts. It may directly inhibit the apoptotic pathway, thus promoting cell survival. By suppressing pro-apoptotic signals and modulating the expression of apoptosis-related proteins, IL-10 can contribute to the reduction in neuronal cell death. In SCI rats, IL-10 overexpression was associated with increased expression of Bcl-2 and Bcl-xL in damaged tissues, initiating an anti-apoptotic cascade and leading to increased neuronal survival and improved motor function [25]. After an SCI, the process of apoptosis is initiated by the mitochondrial secretion of cytochrome c [48]. IL-10 was found to suppress the mitochondrial secretion of cytochrome c in in vitro neurons [25]. Depletion of IL-10 in IL-10 (−/−) mice after an SCI was associated with a significant augmentation of apoptotic cells, reduced Bcl-2 expression, and poorer motor function in comparison to IL-10 wild-type mice [32].

#### 2.2.4. Regulation of Glial Scars

The glial scar is associated with the expression of inhibitory molecules such as CSPGs, which can impede axonal regeneration after an SCI through the initiation of neurite retraction and disruption of the neuronal growth cone [49]. IL-10 may influence the expression of CSPGs, potentially reducing their levels and limiting their inhibitory effects on axonal growth. IL-10 administration decreased CSPG production and suppressed glial scar formation, enhancing neural regeneration and axonal growth [45]. IL-10 expression was augmented when glial scars were removed using Chondroitinase ABC using a p38 MAPK-dependent mechanism [30]. The balance between promoting tissue repair and limiting scar-associated inhibition is a delicate one, and ongoing research aims to elucidate the nuanced effects of IL-10 on scar formation in the spinal cord.

#### 2.2.5. Neurogenic Effects

Studies have demonstrated that IL-10 can promote myelination and axonal regeneration after SCI [39]. In an animal study by Ciciriello et al., IL-10 overexpression promoted spinal progenitors’ survival in a C5 lateral hemisection SCI model [50]. Zhou suggested that IL-10 provides nutritional support to spinal cord neurons via its IL-10R [25]. These neurogenic actions position IL-10 as a promising candidate for therapeutic interventions aimed at preserving and restoring neurological function in an SCI.

#### 2.2.6. Neuroprotective Effects

An SCI results in a transient increase in glutamate, causing secondary injury in the spinal cord [51]. In the CNS, IL-10 suppresses excessive glutamate production, preventing glutamate-induced neurotoxicity [52,53].

#### 2.2.7. Regulation of Vascular Injury

In vivo studies have observed that IL-10 deficiency is associated with increased acute vascular damage after cervical SCI [38].

#### 2.2.8. Reduction in Neuropathic Pain

Neuropathic pain is an important element of the SCI, undermining patients’ quality of life [54]. IL-10 has been observed to reduce neuropathic pain in neurologic SCI models [40]. Zhou et al. observed that IL-10 reduced the phosphorylation of p38 MAPK, decreasing TNF-a production and secretion from microglia in the dorsal horn of the spinal cord, correlating with pain reduction [55].

## 3. Therapeutic Strategies Targeting IL-10 in Spinal Cord Injuries

Given the potential benefits of IL-10 in mitigating the pathological processes associated with SCI, researchers have explored various therapeutic strategies to harness its anti-inflammatory and neuroprotective properties. The majority of studies attempting to evaluate the role of IL-10 in an SCI has been performed on mice and rats, usually adopting incomplete SCI models, including the following types of SCI: contusion with a weight drop device [37,46,56,57] excitotoxic injury with quisqualic acid (QUIS) [40,57,58,59,60], compression with vascular clips [32,61], and lateral hemisection [25].

IL-10 administration in SCI animals has been found to be neuroprotective [40,57,58,59,60], to provide nutritional support to neurons [25], to protect against harmful secondary inflammation by decreasing TNF-α and IL-1β excess [32,37,40], to decrease lesion volume [37,57,58,59], to decrease neuropathic pain [40,58,60], and to improve functional recovery [25,32,56].

### 3.1. Exogenous Administration of IL-10

One approach involves the exogenous administration of IL-10 either by intraperitoneal [37,58,62] or intraspinal injection [40,59]. Studies employing this strategy have reported reduced inflammation, improved tissue preservation, and enhanced functional recovery in experimental models of SCIs. However, challenges such as a short half-life and potential side effects need to be addressed for the successful translation of this approach into clinical settings.

Bethea et al. observed that intraperitoneal administration reduced TNF-α production by spinal cord macrophages and significantly improved functional recovery after SCIs in rats [37]. In IL-10 (−/−) mice, intraperitoneal injection of IL-10 was found to have neuroprotective effects at 1 and 7 days following an SCI [58]. In an animal study by Brewer et al., intraspinal injection of IL-10 led to an increase in SCI lesion volume, while systemic administration resulted in an 18% reduction in lesion volume in SCI rats [59]. Animals receiving systemic injections of IL-10 for 30 min following an experimental SCI showed a significant reduction in neuronal loss within the spinal cord [40]. Systematic administration of IL-10 reduced the longitudinal extent of neuronal loss in the spinal cord in SCI animals [60]. On the other hand, another study of intraperitoneal IL-10 in the treatment of an SCI showed no significant functional recovery [57].

These controversial results may derive from the narrow therapeutic dose of IL-10. In SCI rats, administration of a single dose of IL-10 provoked a significant improvement in locomotor function two weeks after injury, while administration of two doses of IL-10 did not cause any functional recovery [37]. The timing of IL-10 administration is also important. When IL-10 is administered at 6 and 24 h after SCI, it significantly suppresses TNF-α production. No effect was observed when IL-10 was administered at 3 and 7 days after SCI [37].

### 3.2. Gene Therapy

As IL-10 cannot cross an intact blood–cord barrier [63] and has a short half-life [64], systematic administration of IL-10 requires huge doses with potential side effects. Gene therapy represents a promising avenue for sustained IL-10 delivery. The concept behind gene-administered IL-10 is to use genetic engineering techniques to deliver the IL-10 gene directly to the injured spinal cord, allowing for sustained and localized expression of IL-10 and enhancing its therapeutic efficacy [25,56]. This approach aims to harness the anti-inflammatory and neuroprotective properties of IL-10 for therapeutic benefit in the context of SCI [65,66,67,68].

Various vectors, including viral and non-viral, have been employed to deliver IL-10 genes to the injured spinal cord. Viral vectors, such as adeno-associated viruses (AAVs), herpes viruses, or lentiviruses, are commonly used for IL-10 gene delivery [25,39,50,65,67,68,69]. These vectors are engineered to carry the IL-10 gene into target cells, facilitating its integration into the cellular genome or supporting transient expression. An animal study by Chen et al. showed that gene therapy using lentiviral IL-10 can significantly reduce tissue damage and subsequent motor deficits after an SCI [65]. In a C5 lateral hemisection SCI model, IL-10 lentivirus-laden hydrogel tubes increase spinal progenitor survival and neuronal differentiation after an SCI [50]. Delivery of IL-10-encoding lentivirus from multiple-channel bridges reduces neutrophil infiltration and cytokine production and leads to increased numbers of axons and myelination, resulting in improved motor function and neuropathic pain attenuation after an SCI in animal studies [39,68,69]. IL-10 delivered by a nonreplicating herpes simplex virus (HSV)-based gene transfer vector (vIL10) resulted in reduced neuropathic pain and TNF-α expression in rats after an SCI [67]. An animal study by Gal et al. showed that intralesional administration of nucleoside-modified mRNAs encoding human IL-10 resulted in a significant reduction in the microglia/macrophage reaction in the injured spinal segment and induced significant functional recovery [70].

Similarly, in a lateral hemisection injury, the administration of IL-10 through an HSV-based vector increased neuronal survival in the anterior quadrant of the spinal cord and improved motor function up to 6 weeks after an SCI, suggesting that IL-10 may provide direct neuroprotective effects in an SCI [25]. Glutamate levels are increased after an SCI, leading to neurotoxicity. Zhou observed that in vitro neurons treated with glutamate had decreased AKT phosphorylation. Administration of IL-10 restored AKT phosphorylation and substantially improved neuronal survival. By connecting to IL-10R, IL-10 provided neuroprotection against excitotoxicity in vitro through the PI3K-AKT pathway [25]. In addition, it was found that the HSV was an effective vector of IL-10 production in the dorsal root ganglion by reducing mTNF-a nociceptive release after inflammation [55].

### 3.3. Combination Therapies

Combinatorial approaches that target multiple facets of SCI pathology have gained attention. The co-administration of IL-10 with other therapeutic agents, such as anti-inflammatory drugs or growth factors, aims to synergistically enhance neuroprotection and tissue repair. These strategies hold promise for addressing the multifaceted nature of SCIs and maximizing therapeutic outcomes.

In addition to isolated use [40,59], IL-10 has also been used in combination with other therapeutics, like methylprednisolone [57], and transplantation of Schwann cells and olfactory glia [62]. Administration of the combination of methylprednisolone and IL-10 along with Schwann cells or olfactory glia transplantation after moderate contusive SCI of the rat thoracic spinal cord significantly increased the volume of normal-appearing tissue 12 weeks post-injury [62].

In an animal study by Ciciriello et al., IL-10 overexpression provided a friendly microenvironment for cell transplantation therapy and promoted spinal progenitors’ survival in SCI rats [50]. In a C5 lateral hemisection SCI model, IL-10 lentivirus-laden hydrogel tubes increase spinal progenitor survival and neuronal differentiation after an SCI [50]. In an animal study by Gao et al., transplantation of IL-10-overexpressing clinical-grade mesenchymal stromal cells significantly reduced lesion volume, improved axonal regeneration, decreased neuronal apoptosis, preserved neuronal survival, and promoted neuron differentiation in SCI animals [66].

Other bioengineered materials that have been used to provide stable and prolonged secretion of IL-10 include hydrogels [45,50], multiple-channel poly(lactide-*co*-glycolide) bridges [71], and mineral-coated microparticles [33]. Hellenbrand et al. found that mineral-coated microparticles can effectively deliver biologically active IL-10 for an extended period of time, altering macrophage phenotype, attenuating TNF-α production, and aiding in functional recovery after SCI [33]. In a complete transection SCI mouse model, the injected IL-10-releasing hydrogel scaffold decreased proinflammatory cytokine production, enhanced the M2 macrophage/microglia phenotype, and resulted in neural regeneration and axon growth without scar formation [45]. In a mouse SCI model, a bicistronic vector encoding IL-10 was delivered from a poly(lactide-*co*-glycolide) bridge, which provides structural support that guides regeneration, resulting in increased axonal growth, myelination, and subsequent functional recovery [71].

## 4. Discussion

While the potential of IL-10 in SCI treatment is promising, several challenges and unanswered questions persist. The administration of IL-10 is associated with various side effects, as it can have systemic effects on the immune system and other physiological processes. Clinical trials have found that IL-10 can be administered subcutaneously or intravenously at a dose of up to 25 μg/kg without any major side effects [64,72]. At a dose of 100 μg/kg, IL-10 can cause several side effects, including mild-to-moderate flu-like symptoms (fever with chills, headache, fatigue, myalgias, gastrointestinal disturbances, transient lymphopenia, transient thrombocytopenia, elevated transaminase levels, and injection site reactions [64]. High levels of IL-10 may increase the risk of infections by *Listeria monocytogenes* [73], Klebsiella pneumonia [74], and *Streptococcus pneumonia* [75].

Human recombinant IL-10 can be used to mitigate inflammatory responses in various conditions, including autoimmune diseases and inflammatory disorders. Given its ability to suppress inflammatory responses, IL-10 is being investigated as a potential therapeutic agent for autoimmune diseases such as rheumatoid arthritis, psoriasis, inflammatory bowel disease (IBD), and multiple sclerosis. IL-10 may be employed to modulate immune responses in the context of organ transplantation. It could potentially help reduce the risk of graft rejection by suppressing the immune system’s attack on the transplanted organ. IL-10 may be used to regulate the immune response during certain infectious diseases, such as HIV infection and hepatitis C. While an effective immune response is crucial for clearing infections, an overly aggressive response can lead to tissue damage. IL-10 may help balance the immune response to prevent excessive inflammation. In some cases, IL-10 has been investigated for its potential role in cancer immunotherapy. By modulating the immune response, IL-10 may help regulate the body’s ability to recognize and eliminate cancer cells [76,77,78,79,80,81].

At present, there is no clinical study evaluating IL-10 administration in SCI patients. Targeted delivery of IL-10 in animal SCI models has shown significant achievements. While promising results have been observed in preclinical studies, translating these findings into effective clinical therapies requires rigorous testing and validation. Bridging the gap between preclinical research and clinical application poses challenges. The heterogeneity of SCIs, the optimal dosing and timing of IL-10 administration, and the potential for adverse effects necessitate further investigation. Additionally, understanding the interplay between IL-10 and other molecular players in the complex SCI microenvironment will be crucial for the development of targeted and effective therapeutic interventions. However, several important considerations need to be addressed through further research, such as delivery to the injury site, blood–brain barrier permeability, optimal dosage and timing, risk of systemic side effects, short half-life, bioavailability, and stability. As IL-10 administration increases the risk for bacterial infections, taking into consideration that the main causes of death in SCI patients have been septicemia and pneumonia, treatment of these patients with IL-10 may be dangerous without careful monitoring and appropriate antimicrobial therapy.

Individual patients may respond differently to IL-10 treatment based on factors such as the severity of the injury, the stage of the SCI, and the patient’s overall health. Personalizing treatment plans to account for these variables is challenging. Understanding the long-term effects of IL-10 administration and establishing appropriate follow-up protocols are essential. Monitoring patients over extended periods is crucial to assessing the sustained therapeutic effects and potential late-onset complications.

## 5. Conclusions

In conclusion, IL-10 emerges as a pivotal player in the pathogenesis and treatment of SCI. Its multifaceted role in modulating inflammation, oxidative stress, neuronal apoptosis, glial scars, and neuroprotection positions IL-10 as a promising therapeutic target. The ongoing research exploring various strategies for harnessing the potential of IL-10 offers hope for the development of effective treatments that could significantly improve outcomes for individuals suffering from spinal cord injuries. As our understanding of IL-10’s intricacies deepens, it opens new avenues for innovative and targeted therapeutic interventions, bringing us closer to the goal of alleviating the profound impact of SCI on patients’ lives.

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
