# Peer review of "The Role of Interleukin-10 in the Pathogenesis and Treatment of a Spinal Cord Injury"

_diagnostics, 2024, doi:10.3390/diagnostics14020151_

Round 1
Reviewer 1 Report
Comments and Suggestions for Authors
Although not a hot topic with regard to the role of IL-10 in spinal cord injury and its therapeutic effects, this review contains a great deal of information on IL-10 in spinal cord injury and should be a useful article for readers working in similar areas.
It is well organized in the main, but there are a few points that need to be revise.
1. Lines 92-101
The introduction is somewhat redundant. The role of IL-10 in diseases other than spinal cord injury may not be important to this review.
2. There are several typos.
Line 18: administation
Line 20 regeneraton
Line 271: functiona
Lines 329-330: monocytogenes monocytogenes
Author Response
Dear Reviewer 1,
The paragraph in the introduction stating the role of IL-10 in other conditions has been reduced. The typos mentioned, as long as a few more have been corrected.
Thank you very much for your time and cooperation.
Reviewer 2 Report
Comments and Suggestions for Authors
In this review article, the authors examined the role and possible molecular mechanisms of Interleukin-10 in SCI. The language is clear and concise and the literature cited are accurate and abundant. However, there are several issues that need to be addressed in the manuscript.
1) In page 4, the authors discussed the implications of microglia/macrophage polarization in secondary injuries induced by SCI. The nomenclature of M1/M2 is considered by many in the neuroinflammation field to be outdated. See the article entitled: Abandoning M1/M2 for a Network Model of Macrophage Function.
2) In the section regarding the role of IL-10 in the pathogenesis of SCI, a few subsections only have two or three sentences. Is it necessary to divide the topic into these smaller subsections? Furthermore, is it necessary to include subsections where the literature is so scant? The authors should either try to expand on these subjects or drop them entirely.
Author Response
Dear reviewer 2,
1) The M1/M2 nomenclature was replaced by the newest data on macrophage/microglia categorization as indicated
2) Dividing the paragraph into subsections is better for readers as it makes it more clear and easier to understand. Furthermore, we think that even a little information regarding the mentioned subtopics must be included in our work. On the other hand, trying to expand its subsection further does not fit our work's scope.
Thank you very much for your time and cooperation.